# Measuring the Spatial Allocation Rationality of Service Facilities of Residential Areas Based on Internet Map and Location-Based Service Data

**Xinxin Zhou** [1,2], **Yuan Ding** [3], **Changbin Wu** [1,2,4,*], **Jing Huang** [1,2] and **Chendi Hu** [1,2]

1   College of Geographical Science, Nanjing Normal University, Nanjing 210023, China;
    windofmay@foxmail.com (X.Z.); huangjing04030@163.com (J.H.); 15295576608@163.com (C.H.)
2   Key Laboratory of Virtual Geographic Environment, Ministry of Education, Nanjing Normal University,
    Nanjing 210023, China
3   School of Earth Science and Engineering, Hohai University, Nanjing 211100, China;
    dingyuanhhu@hhu.edu.cn
4   Jiangsu Provincial Key Laboratory for Numerical Simulation of Large Scale Complex Systems,
    Nanjing Normal University, Nanjing 210023, China
*   Correspondence: wuchangbin@njnu.edu.cn

**Abstract:** The spatial allocation rationality of the service facilities of residential areas, which is affected by the scope of the population and the capacity of service facilities, is meaningful for harmonious urban development. The growth of the internet, especially Internet map and location-based service (LBS) data, provides micro-scale knowledge about residential areas. The purpose is to characterize the spatial allocation rationality of the service facilities of residential areas from Internet map and LBS data. An Internet map provides exact geographical data (e.g., points of interests (POI)) and stronger route planning analysis capability through an application programming interface (API) (e.g., route planning API). Meanwhile, LBS data collected from mobile equipment afford detailed population distribution values. Firstly, we defined the category system of service facilities and calculated the available service facilities capacity of residential areas (ASFC-RA) through a scrappy algorithm integrated with the modified cumulative opportunity measure model. Secondly, we used Thiessen polygon spatial subdivision to gain the population distribution capacity of residential areas (PDC-RA) from Tencent LBS data at the representative moment. Thirdly, we measured the spatial allocation rationality of service facilities of residential areas (SARSF-RA) by combining ASFC-RA and PDC-RA. In this case, a trial strip census, consisting of serval urban residential areas from Wuxi City, Jiangsu Province, is selected as research area. Residential areas have been grouped within several ranges according to their SARSF-RA values. Different residential areas belong to different groups, even if they are spatially contiguous. Spatial locations and other investigation information coordinate with these differences. Those results show that the method that we proposed can express the micro-spatial allocation rationality of different residential areas dramatically, which provide a new data lens for various researchers and applications, such as urban residential areas planning and service facilities allocation.

**Keywords:** service facilities; Tencent location-based data; points of interest (POI); spatial allocation; internet map

## 1. Introduction

The provision of amenities/service facilities is one of the issues of concern regarding urban inequalities in social science research [1]. Public property attributes ensure the equivalence operation of

civic social and economic activities [2]. Several guiding documents define service facilities. For example, "Urban Residential Area Planning and Design Specification (GB50180-93, version 2002)", which is one of the Chinese urban planning national standards, defines service facilities as those that meet the basic needs of life and are compatible with the size of the population [3]. The policy of shifting the priority of urban construction and public service resources allocation to the suburbs in Shanghai was outlined by a government work report in 2019, which profoundly explained the city government's policy orientation to revitalize the disadvantaged suburbs. The planning target of spatial allocation rationality of service facilities of residential areas (SARSF-RA) is to achieve equivalence between the accessible service facilities of residential areas and the population of residential areas [1,4,5]. Through a geographer's lens, SARSF-RA becomes complicated and increasingly microscopic due to the combined influence of natural and human factors, especially in China and other countries experiencing rapid development [6]. Therefore, it is desiderated to adopt new, representative datasets that are good at the micro-spatial scale to calculate the spatial allocation rationality of the service facilities of residential areas.

The study of the spatial accessibility of service facilities is a precondition for SARSF-RA and urban evenness measurement [7]. The accessibility of service facilities research continues to receive attention [8,9]. Previous scholars have mainly focused on a single type of service facilities when examining accessibility, such as public healthcare services [8,10], urban green space [11], commercial facilities [12], or transit services [13]. These accessibility evaluations can identify the potential shortcomings of typical services and reflect the balance of facilities' spatial layout. However, few comprehensive assessments of the accessibility of various service facilities have been researched [14], and the impact of diversity on the accessibility of service facilities has not been revealed.

There are two indicators for measuring the accessibility of service facilities: the travel costs and the quality/quantity of opportunities [15]. The travel costs represent the actual ability to reach the provider through different transportation systems [16], in which travel distance or travel time is considered regular. The quality/quantity of opportunities contains different aspects from different levels, such as population flow from a macro level and population count from the micro level [17]. The traditional measurement for travel costs is mainly based on the Geographical Information System (GIS) analysis of origin–destination matrices, which need valuable fundamental transportation network data [13]. The conventional data source that is used to measure opportunities is often statistical survey data. There are apparent deficiencies for traditional data, such as relying on the intensive investigation, which is costly, time-consuming, and too coarse for detailed mapping and small areas. It is challenging for this situation to meet the requirements of high efficiency and a microcosmic scale [18].

The pervasive adoption and penetration of new research data, such as Internet map and location-based service (LBS) data, create an upcoming research stream, which retrieves detailed information about local amenities [19]. Internet mapping services, which are also called web-based mapping services, such as Google Maps (https://www.maps.google.com), Amap (https://www.amap.com), Baidu Map (https//www.baidu.map.com), are used to acquire the travel costs and complete point of interest (POI) objects directly based on a route planning application programming interface (API) and POI API [17,20]. (1) Route planning APIs consider different modes of transport, such as driving, walking, cycling, and public transport [13,21]. It is feasible to measure the travel time costs through route planning API [22,23]. (2) POI data, a burgeoning data source for regional research, is applied for the automated identification and characterization of parcels (AICP) [24], extracting hierarchical landmarks [25], and the hierarchical semantic cognition of urban functional zones [26,27]. POI data can estimate employment locations, and implement interzonal commuting patterns [20]. (3) LBS data contain information on people's hobbies and places, and have provided new informat ion in terms of understanding individual activity patterns [28]. Representative LBS data include Twitter data and Weibo data, which can be used for dynamic land-use mapping [28] and assessing human travel activity patterns [29]. Tencent LBS data (https://heat.qq.com) are quintessential data that record more than 50 billion location data of dominant Chinese from China's largest social media operator [30]. Tencent LBS data provide rich user information, including population location data and population

category descriptions, which afford more microcosmic, dynamic, and real population data for assessing SARSF-RA. Thus, we can use POI data and LBS data as the data of the quality/quantity of opportunities.

Based on the above description, Internet map and LBS data that provide micro-scale knowledge about residential areas will be pertinent to SARSF-RA. In the next section, we will introduce and analyze detail data information and the measurement framework.

## 2. Materials and Methods

### 2.1. Study Area and Data

This paper selected a trail strip census in Binhu District, Wuxi City as the experimental area (Figure 1). Wuxi is a typical city in the Yangtze River Delta region of China. It ranked 18th among the top 100 cities in China, with 10.5118 trillion gross domestic product (GDP) and 6,553,300 residents in 2017. Binhu District is dominated by secondary and tertiary industries, among which the township of Hudai has a tax sale of 20.107 billion. BinHu District was rated as one of the top 100 areas with investment potential in China in 2018, in which representative industries include the Internet of Things (IoT), Computer Aided Industry Design (CAID), software service outsourcing, and so on. The experimental area is located in the Wuxi Hudai Industrial Zone, which borders Taihu Lake in the southeast. It is the only peripheral town in BinHu District, and is thus relatively independent of the core urban area, where the population is dense, and the industry mainly relies on electronics, light, and textiles. The experimental area was used to research the carrying capacity pressure of the service facilities contained 14 communities and 13,600 residents. This paper collected Internet map data (route planning API, POI data, and remote sensing images) and LBS data (Tencent LBS data). A detailed introduction about the data volume, data descriptions, and data applications can be found in Table 1.

**Table 1.** A detailed description of the data used in the paper. API: application programming interface, LBS: location-based service, POI: point of interest.

| Data Category | Dataset Name | Data Content | Data Application | Access Method |
|---|---|---|---|---|
| Internet map data | Route planning API | Three route planning methods, included driving, riding, and walking, requested opening API to obtain corresponding planned path. | Calculating travel time costs | Requesting from https://lbs.amap.com |
| | POI data | POI data was requested from Amap in BinHu District in 2018, including schools, banks, and restaurants. | Representing different service facilities | Requesting from https://lbs.amap.com |
| | Basic geographic data | Amap platform provides remote sensing image to request. | As map background | Downloading from https://ditu.amap.com |
| LBS data | Tencent LBS data | Requesting real-time population locations of Tencent one time per hour from 9 April 2018 to 22 April 2018; attained 1,570,000 records. | Reflecting spatial distribution of real population | Crawling from https://heat.qq.com/ |

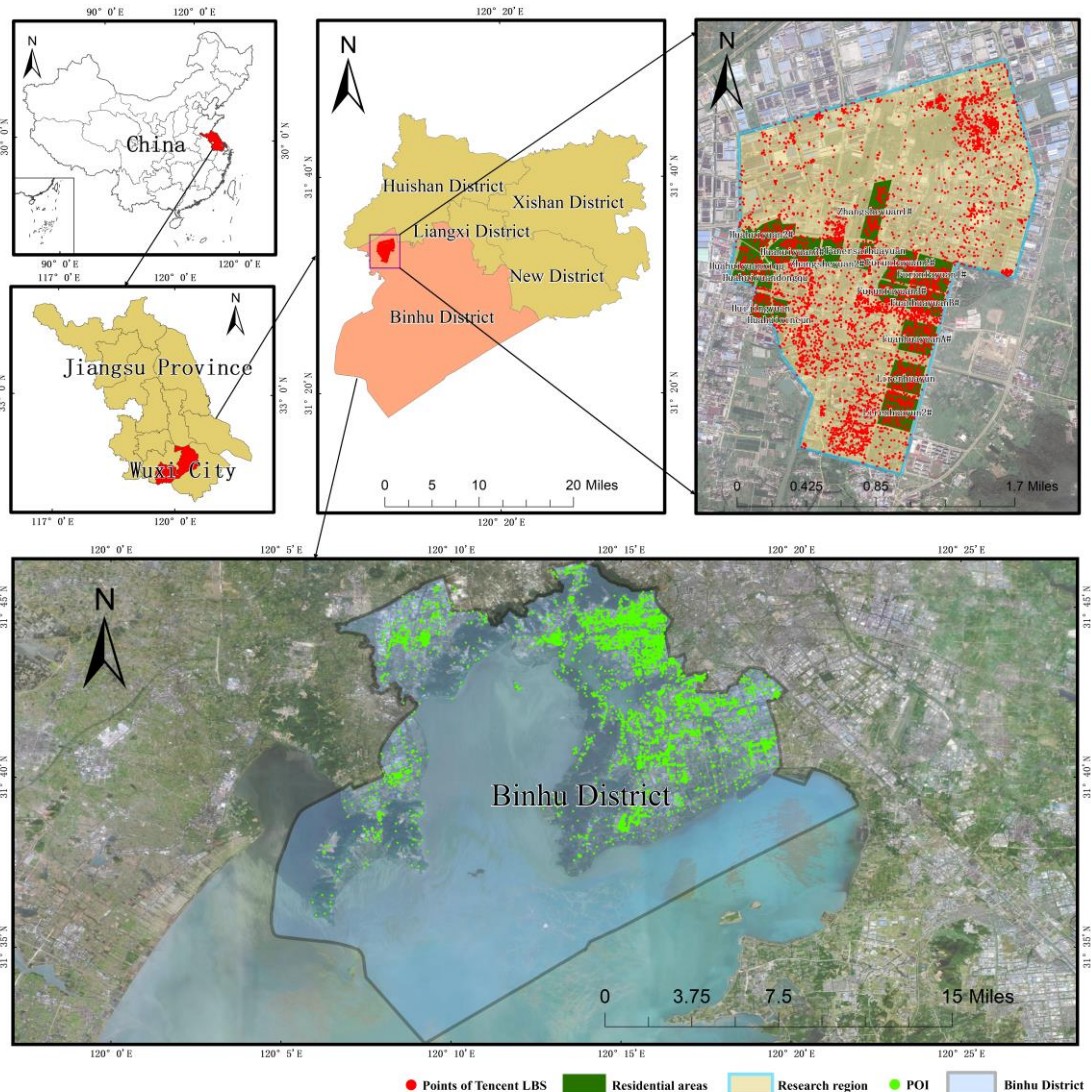

**Figure 1.** The research area: the urban built-up area of Hudai Town. The red dots represent the distribution of the point of interest (POI), the green blocks represent the researching community, and the light yellow blocks represent the research region.

## 2.2. The Assessment Methodology

In general, the framework of assessing SARSF-RA has six steps (Figure 2): (I) basic data preparation based on three crawler programs to crawl POI data, Tencent LBS data, and basic geographic data; (II) building the category system of service facilities and defining the travel time costs threshold; (III) gaining the travel time costs to service facilities using the route planning API program; (IV) calculating the available service facilities capacity of residential areas (ASFC-RA) through a modified cumulative opportunity measure model; (V) building the population distribution capacity of residential areas (PDC-RA) based on Tencent LBS data representative moments selection and the Theisson polygon space segmentation method; and lastly, (VI) measuring the SARSF-RA values and analyzing the regularity and characteristics of different residential areas.

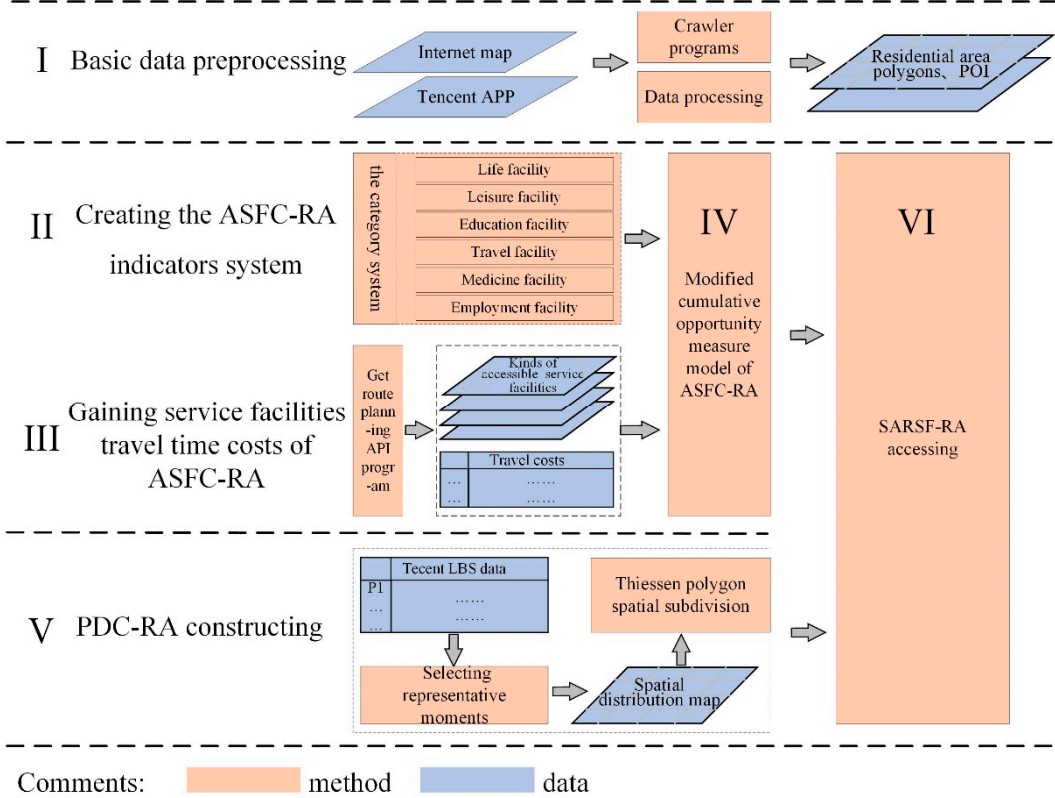

**Figure 2.** Overall assessment roadmap.

*2.3. The Construction Process of ASFC-RA*

We will introduce the construction process of ASFC-RA, which consists of steps II, III, and IV in Figure 2. The premise of diversification is a feature of a reasonable and comprehensive classification system of service facilities; thus, the calculation travel time costs based on route planning API from Internet map data is a convenient spatial standard for determining whether the service facilities are accessible or not. In the following sections, we outline the design philosophy of a modified cumulative opportunity measure model through various types of service facilities.

(1) The category system of service facilities

Firstly, it is necessary to construct a service facilities classification system in the residential areas. The classification system of service facilities in residential areas will be built based on the four concepts of basic human living requirements proposed by the World Health Organization. First, the system should be selected according to the principle of ensuring living conditions, and this information then needs to be combined with the living environment of the city [31]. This paper refers to a total of 339 indicators in the evaluation index systems of nine livable cities at home and abroad; these indicators comprehensively consider the attributes of residential life services in residential areas from six aspects, including life services, education services, leisure services, transportation services, medical services, and employment services. The detail category information is as shown in Table 2.

The identified methods of travel include walking, public transportation, self-driving, cycling, and combinations of these methods [22]. Many factors influence the choice of travel mode, including economic conditions, travel distance, and travel urgency. Walking is the most basic and most common mode of travel for human beings [32]. The principle that is usually followed is as follows: people will choose to walk if a destination can be easily reached by walking. Therefore, to quantitatively judge the service radius of service facilities, the walking mode is used for measurement purposes. To determine the passing time of the travel time costs threshold, the time cost of five types of service

facilities, including life, education, leisure, travel, and medicine, was determined to be 30 minutes. It is believed that employment causes a specific type of commute between work and home, and this commuting distance can be significant and differs from that of other trips; thus, the time was selected as 60 minutes.

**Table 2.** The category system of service facilities.

| Classification | Service Name | Indicator Interpretation |
|---|---|---|
| Life | life service | Life service place, travel agency, information consultation center, ticket office, post office, express delivery, telecommunication business hall, office, water supply business hall, and electric power business hall |
| | shopping service | Shopping malls, convenience stores, home appliance electronics stores, supermarkets, home building materials markets, stationery stores, sports stores, shoes, hats and leather stores, and personal products/cosmetic stores |
| | catering service | Catering-related places, Chinese restaurants, foreign restaurants, fast food restaurants, casual restaurant, cafes, tea houses, cold drink shops, pastry shops, and dessert shops |
| | accommodation service | Accommodation services, hotels, and hotel guest houses |
| | financial insurance service | Financial and insurance services, banks, automated teller machines (ATMs), insurance companies, securities companies, and finance companies |
| | public utilities | Public toilets, funded shelters, service facilities, newsstands, and public telephones |
| | business residence | Related business housing and residential areas |
| Education | science and culture service | Science and culture education sites, museums, convention centers, art galleries, libraries, science and technology museums, planetariums, cultural palaces, literary and art groups, media organizations, schools, research institutions, and training institutions |
| Leisure | park facility | Comprehensive parks, zoos, botanical gardens, children's parks, and gardens providing places for residents to enjoy, watch, relax, and enjoy scenic spots |
| | sports and leisure services | Sports and leisure service places, sports venues, entertainment venues, resorts, leisure venues, and theaters |
| Travel | parking lot and repair facility | Gas stations, car sales, car repairs, private and public parking lots, parking spaces, auto repair shops, automobile sales service shop |
| | transportation facilities service | Related airport, railway station, long-distance bus station, subway station, light rail station, bus station, shuttle bus station, parking lot, border port, taxi, ferry station, and ropeway station |
| | road auxiliary facilities | Road auxiliary facilities, warning information, toll stations, service areas, traffic lights, and street signs |
| Medicine | medical facility | It mainly includes first-level, second-level, and third-level hospitals, community clinics, private clinics, private hospitals, pharmacies, general hospitals, specialist hospitals, and emergency centers |
| Employment | public enterprise | Companies, factories, bases with agriculture, forestry, herds, and fish |

(2) Gaining accessible service facilities of the ASFC-RA

This paper uses Python3.6 and ArcGIS ArcPy (10.2, Esri, Redlands, California, America) to write a parsing program for layer data parsing to gain the accessible residential service facilities through route planning API from Amap. The code has been uploaded to GitHub, and the address is https://github.com/windofmay5/PublicFacilitiesNavigation. The specific steps are as follows:

(1) Data input: Respectively input the residential area polygons layer and the POI service facilities layers, which are multi-layered, where the residential area polygons layers are used as the starting list, and the POI service facilities layers are used as the ending list.

(2) Service radius input: Enter the travel time costs threshold identification list as the criteria for judging the threshold of reachable service facilities.

(3) Traversing the center points of residential area polygons and service facilities to verify its travel time: Clarify the start points and end points, and then request Amap path planning API for the dynamic planning of walking distance time. By splicing the HyperText Transfer Protocol (HTTP), request the Uniform Resource Locator (URL) to receive the data (JavaScript Object Notation(JSON) or Extensible Markup Language(XML) format) returned by the HTTP request, then parse the data, e.g.,

http://restAPI.amap.com/v3/direction/walking?origin=116.45925,39.910031&destination=116.587922,40.081577&output=xml&key=<thekeyofusers>.

(4) Dictionary output: Output the number of different types of service facilities in each district. The specific implementation process is the following pseudo-code.

---

**Program 1:** Gaining service facilities of available service facilities capacity of residential areas (ASFC-RA) through Amap route planning API

---

| | |
|---|---|
| **1** | ***Input:** the POI service facilities layer of one experimental area is a list of layer_s; the residential areas layer names layer_D* |
| **2** | ***Input:** the travel time costs threshold identification list is T=[t1,t2, … … tS]* |
| **3** | ***Output:** a two-dimensional array of different types of service facilities in all the residential areas is named accessArray[D, S]* |
| **4** | *D = count of residential areas* |
| **5** | *S = count of service facilities types* |
| **6** | ***for** s = 0; s < S; s++ **do*** |
| **7** | *#Find out the service facilities of corresponding communities in S types of facilities* |
| **8** | *#Get the s service facilities layer deposited in layer_s* |
| **9** | ***For** j = 0; j < length(layer_s); j++ **do*** |
| **10** | *#Get the long-lat of the service facilities point j and save them into the facility* |
| **11** | ***For** I = 0; I < D; i++ **do*** |
| **12** | *#Get the longitude and latitude of the community i and store them into the residential area* |
| **13** | *#Request route planning API, return JSON object and store result, the request form is: #request.url(http://restAPI.amap.com/v3/direction/walking?origin=facility.X,#facility.Y&destination=residentialarea.X,residential#area.Y&output=json&key=<thekeyofusers>)* |
| **14** | *#get the current path planning time in the result and store it into timeIJ* |
| **15** | ***if** timeIJ<T[s] **then** ## service facilities travel time costs* |
| **16** | *#Update the row i of the accessArray, and the s column object counts, accessArray[i, s]++* |
| **17** | ***return** accessArray* |

---

Using the program of gaining service facilities of ASFC-RA by Amap route planning API, the number of various service facilities POIs within the travel costs threshold is obtained. The results are listed in Table A1.

(3)　The modified cumulative opportunity measure model

The accessibility method of various service facilities is mainly measured by the number of service facilities and the variety of service facilities [33]. This paper uses the cumulative opportunity method to estimate the number of different service facilities in different residential areas. The approach to

measure the number of various service facilities POIs within the travel costs threshold is intuitive and attractive to reflect the number of service facilities available for a given residential area.

$$V_{ASFC-RA}(i,s,t) = \sum_j G(t_{ij}) \times F_j \tag{1}$$

In Equation (1), $V_{ASFC-RA}(i,s,d)$ is the accessibility value of a type $s$ service facility for a residential area $i$ in a certain travel time costs threshold $t$. $G(t_{ij})$ is a binary variable, when $t_{ij}$ within a certain travel time costs threshold $t$, it is one; otherwise, it is zero. $t_{ij}$ is the actual route planning travel time cost between residential area $i$ and service facility $j$. $F_j$ is the opportunities for service facility $j$, which can be the economic scale or building area of service facility $j$ commonly. In consideration of there being no uniform scale of POI data, we set $F_j$ to one.

$$V'_{ASFC-RA}(i) = \sum_s V_{ASFC-RA}(i,s,t) \tag{2}$$

$V'_{ASFC-RA}(i)$ in Equation (2) is the accessibility value of all types of service facilities for residential area $i$. $V'_{ASFC-RA}(i)$ reflects the quantity obtained by all service facilities, but it does not reflect the diversity of service facilities. Based on the existing research methods, the diversity of the service facility can be measured from the Shannon entropy index, which is shown as Equation (3):

$$H_i = -\sum_S [P_i(s) \times ln P_i(s)] \text{among them, } P_i(s) = V_{ASFC-RA}(i,s,t)/V'_{ASFC-RA}(i) \tag{3}$$

In Equation (3), $H_i$ is the Shannon entropy of residential area $i$, while $P_i(s)$ is the ratio between $V_{ASFC-RA}(i,s,t)$ and $V'_{ASFC-RA}(i)$.

$$W_i = H_i/lnS \tag{4}$$

Based on $H_i$, the relative entropy $W_i$ is the relative entropy weight in Formula (4), which ranges from zero to $lnS$. $S$ is the species number of service facilities. Multiple types of service facilities can have different effects on different residential areas. The modified cumulative opportunity measure formula based on the entropy weight index is shown in Equation (5):

$$V''_{ASPF-RA}(i) = \sum_s V_{ASFC-RA}(i,s,d) \times W_i \tag{5}$$

Normalization that transforms the dimensional expression into dimensionless expression is a way to simplify the calculation process. The values of ASFC-RA that have been max–min normalized is obtained by the normalization Equation (6). $V''_{ASPF-RA}(max)$ indicates the maximum value of $V''_{ASPF-RA}(i)$, and $V''_{ASPF-RA}(min)$ indicates the minimum value of $V''_{ASPF-RA}(i)$.

$$V_{ASFC-RA}(i) = \frac{V''_{ASPF-RA}(i) - V''_{ASPF-RA}(min)}{V''_{ASPF-RA}(max) - V''_{ASPF-RA}(min)} \tag{6}$$

### 2.4. The Implementation Process of PDC-RA

We will introduce the implementation of PDC-RA based on Tencent LBS data. Considering that the research area's population size changes over the course of a day, we should select a representative moment. We calculate the PDC-RA value by the Thiessen polygon spatial subdivision method.

(1)　Choosing a representative moment

Population size data at different hours in multi-days were obtained from Tencent's location-based big data platform. We analyze the temporal and spatial variation and pick a representative moment Tencent LBS data as the data for calculating PDC-RA. Through 24 hours of group summation statistics, the temporal change characteristics of the population quantity in the experimental area during multiple

days (weekdays and weekends) are shown in Figure 3. Weekdays and weekends show the same trend of change. The lowest population was at approximately 05:00, and the peak population occurred at about 21:00. What's more, 94% of people are at home according to a national time-use survey [34]. We select the 21:00 population points data as the research area's actual carrying population. The 21:00 population points data are processed to a 21:00 population points layer: 21PPL for short.

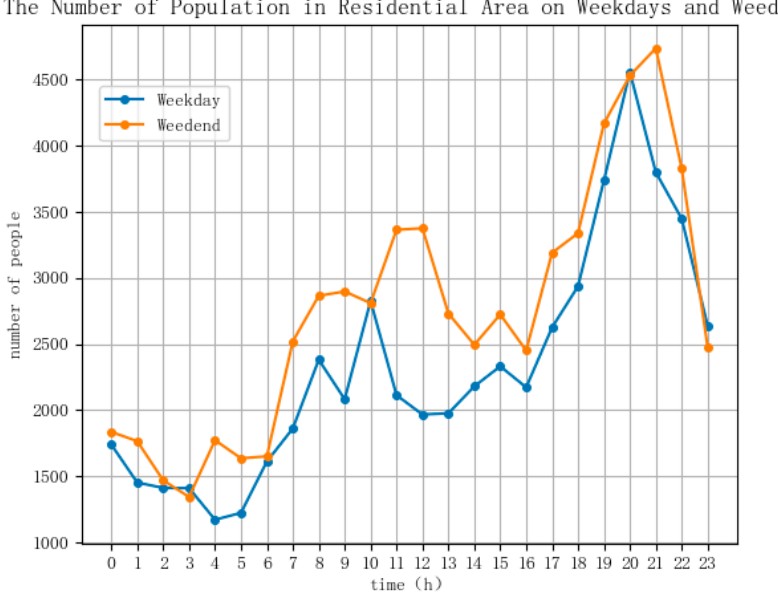

**Figure 3.** Comparison diagram of population size changes over a weekday and weekend.

(2)  Thiessen polygon spatial subdivision

Each point object population value of 21PPL represents the statistical aggregation degree of its circumjacent coverage, which is calculated from Tencent's location-based big data platform. If we intersect points of 21PPL with residential areas, there will be a great spatial error. So, studying the method for mapping discrete points into continuous space is the basis for counting the population of residential areas. We adopt the characteristics of "any position in the polygon is closest to the sample point" in the Thiessen polygon, construct the population distribution statistical point data using the Thiessen polygon on Tencent LBS data, and apply the spatial association rules by intersecting with plots, as shown in Figure 4. Thiessen polygons are a type of partition of the space plane. The method is characterized by any position within the polygon being the closest to potential points. Each polygon contains only one sample point. In considering of the equal division characteristics of the Thiessen polygon in spatial segmentation, the method can be used to solve problems such as nearest points, minimum closed circles, spatial analyses, adjacency, proximity, and reachability analyses.

After the points of 21PPL are allocated to each residential area, the population size of the residential area is calculated. The calculation equation of the population in each residential area is as shown in Equation (7):

$$V'_{PDC-RA}(i) = \sum_{r=1}^{u} \left( \frac{A_c(r)}{A(r)} \times P(r) \right)(r = 1, 2 \ldots, u) \tag{7}$$

$V'_{PDC-RA}(i)$ is the PDC-RA value of residential area $i$. $u$ indicates the number of Thiessen polygons that are superimposed on residential area $i$, $A_c(r)$ indicates that the sub-block area of residential area $i$ intersects with the Thiessen polygon $r$, $A(r)$ indicates the area of the Thiessen polygon $r$, and $P(r)$ indicates the population value of the Thiessen polygon $r$. The PDC-RA results are shown in Table A1.

The normalization Equation (8) gains the values of PDC-RA that have been max normalized. $V'_{PDC-RA}(max)$ indicates the maximum value of $V'_{PDC-RA}(i)$. The purpose of max normalization is to prevent PDC-RA values from reaching zero and not being applied to SARSF-RA calculations.

$$V_{PDC-RA}(i) = \frac{V'_{PDC-RA}(i)}{V'_{PDC-RA}(max)} \tag{8}$$

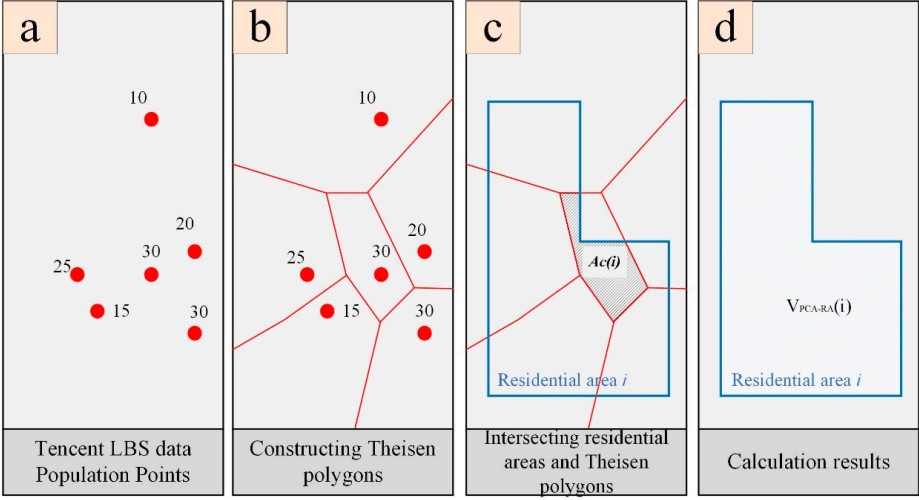

**Figure 4.** Calculation diagram of population distribution.

## 2.5. The Measurement of SARSF-RA

The fundamental meaning of SARSF-RA is the coordination ratio of the value of ASFC-RA and PDC-RA. The SARSF-RA values are dimensionless scalar, which directly reflects the spatial allocation rationality. As the SARSF-RA value increases, the spatial allocation rationality of service facilities of residential areas become high. The $V_{SARSF-RA}(i)$ in Equation (9) indicates the SARSF-RA value of residential area $i$.

$$V_{SARSF-RA}(i) = \frac{V_{ASFC-RA}(i)}{V_{PDC-RA}(i)} \tag{9}$$

## 3. Results

### 3.1. The ASFC-RA Results

We calculated the accessible service facilities level of various facilities in each residential area, and these are shown as the results of the indicator calculations in the experimental area in Table A1. The difference regarding the number of accessible service facilities of residential areas were prominent, and the spatial distribution of different types of service facilities is shown in Figure 5. The basic characteristics of the resulting data are as follows. (1) The quantity (Q) order of accessible service facilities in different residential areas showed a basically consistent order: Q(employment services) > Q(life services) > Q(travel services) > Q(leisure services) > Q(medicine services) > Q(education services), but some residential areas have variations; for example, Q(medicine services) in HuiJingYuan is slightly higher than Q(leisure services). (2) Q(employment services) is the most significant number, which is far higher than the number of any other service facilities, and each of the Q(employment services)s is more extensive than 1400. Furthermore, the main influencing factor is that the time standard of the travel time costs threshold for enterprise service facilities is 60 minutes, which is greater than the travel time costs threshold of the other service facilities. Also, compared with field data, the experimental area is located in the Wuxi Hudai Industrial Zone. There is a total planned area of 10.3 km$^2$ in the Wuxi Hudai Industrial Zone. It is one of the essential industrial zones in Wuxi City; it integrates industry, commerce, and logistics, and focuses on electronics, light,

textiles, machinery, and logistics. The number of enterprise facilities, 1642, is the largest in Huahui Yuan #2, and the community is adjacent to the Wuxi Hudai Industrial Zone. (3) The values of Q(life services) are also high, and each value is greater than 800. The reason for these large numbers is that Hudai is located in the Binhu District of Wuxi, and has developed economies and various living and service facilities. The residential area with the lowest number of living facilities is the LiRen Garden #2 community, which is a new residential area that was established in 2017. The personnel occupancy is small, and the living service facilities in the surrounding areas are limited. (4) The values of Q(travel services), Q(leisure services), Q(medicine services), and Q(education services) are less than 200, which is related to the service attributes provided by the service facilities themselves, e.g., the medical services, travel services, education services, and leisure services. Most of the educational facilities belong to social welfare service facilities, which are generally undertaken by the government, and the number of people served by each service facility is high, e.g., educational facilities and sports facilities, primary schools, middle schools, and stadiums; furthermore, these facilities have a larger scale and a wide range of services.

To better understand the characteristics of the spatial distribution of accessible service facilities, we designed six maps (Figure 6) of different service facilities to express the quantitative difference between mapped features by varying the color of symbols. We found the location of the high-value or low-value elements to be clustered in space. In general, all types of accessible service facilities maps have a similar spatial distribution pattern that shows high-value aggregation in the northwestern direction and low-value aggregation in the southeast direction. The high-value group included HuaHuiXinCun, HuaHuiXinCun#1, HuaHuiXinCun#2, HuaHuiXinCun#3, HuiJingYuan, and HuaHuiYuan; the mid-value group included Versailles Estate, ZhangSheYuan#1, FuRunHuaYuan#3, FuRunHuaYuan#2, FuRunHuaYuan#1, FuAnHuaYuan #B, and FuAnHuaYuan #A; and the low-value group included LiRen Garden#2 and LiRen Garden#1. This phenomenon certifies that the overall service resource supply level was relatively high in the northwestern region of the experimental area, and the service resource supply in the southeast area was relatively infertile. The spatial distribution map of ASFC-RA is shown in Figure A1a. The high-value region is over HuaHuiYuan, HuiJingYuan, HuaHuiXinCun, and HuaHuiYuan#2, too, which reflects that this region has comprehensive and intensive service facilities. Through field surveying, this region is located at the central part of Hudai Town, which intensifies two factors; facilities quantity and travel times certify the ASFC-RA reasonableness.

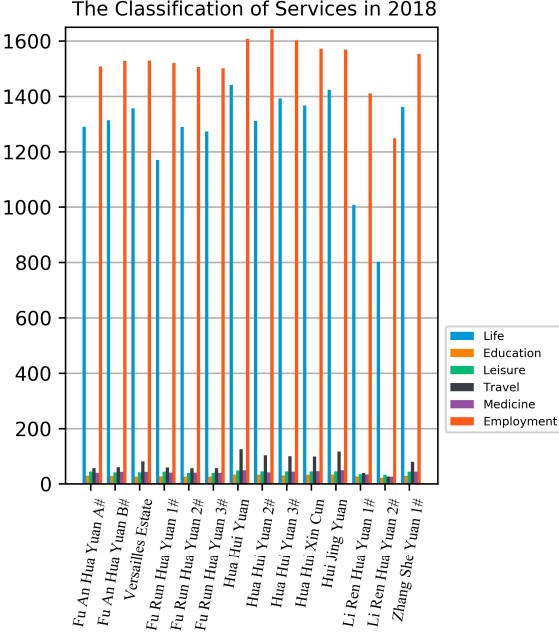

**Figure 5.** Bar chart of the number of various types of service facilities in residential areas.

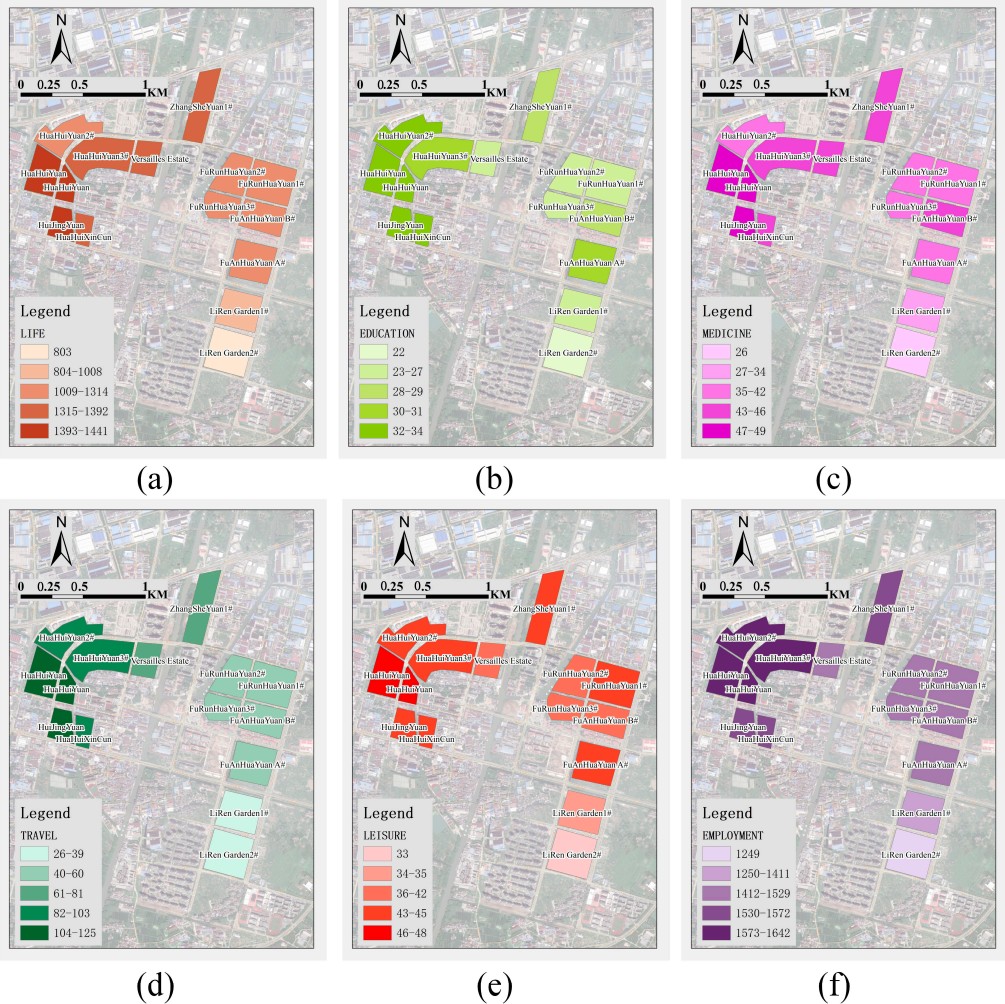

**Figure 6.** Graduated color maps of various types of service facilities in residential areas.

## 3.2. The PDC-RA Results

The spatialized expression of PDC-RA values in the experimental area is shown in Figure A1b, and the bar charts after the quantitative expression are shown in Figure A2b. The PDC-RA values are different among residential areas. To further reveal the characteristics of each residential area, the basic building attributes of the residential area were collected from a real estate agency website, and these data included the total building area, makespan, and average building floors, as shown in Table A1. The following properties can be summarized. (1) The larger number of inhabitants was found in FuAnHuaYuan #B and LiRen Garden#1, where the populations were 2904 and 2888, respectively. By analyzing the basic building attributes, the building average floors in the residential areas were greater than 10, and the building area values were 105,389 m$^2$ and 365,801 m$^2$. Thus, FuAnHuaYuan #B was more densely populated than LiRen Garden#1. (2) The lower population was found in Versailles Estate, which was 376. The residential area is a high-end villa residential area with three floors and building area of 10,169 m$^2$. For verifying the validity of the Tencent LBS data and the implementation process of PDC-RA in this paper, the average building area per person was approximately 43 m$^2$ (excluding extreme values, i.e., FuAnHuaYuan #B, LiRen Garden#1, and Versailles Estate). The 2016 Jiangsu Statistical Yearbook records indicated that the housing area per person was 47 m$^2$. The two indicators are close to each other. Thus, the value aligns with the real situation, and further verifies the effectiveness of Tencent LBS data and the methods reported in this paper.

*3.3. The SARSF-RA Results*

The SARSF-RA result is as shown in Figure A2c, and the spatial distribution map is as shown in Figure A1c. It was concluded that: (1) the highest SARSF-RA value was in Versailles Estate, i.e., 4.804; the SARSF-RA of LiRen Garden#2 was the smallest at zero, and this value was related to the processing method of the ASFC-RA standardized data. Thus, the Versailles Estate residential area had the strongest comprehensive service capability and the highest value of reasonableness. (2) The SARSF-RA values of HuaHuiXinCun and HuiJingYuan were higher, at 2.750 and 2.680, respectively. It is easy to determine from the spatial distribution that the two residential areas are adjacent to each other and are located in the commercial area. Also, these two communities are low-floor mature communities that were built before 2010. The overall service capacity of the low-floor mature residential area located in the commercial center was ranked second. The SARSF-RA values in LiRen Garden#1 and LiRen Garden#2 were the lowest, at 0.280 and 0, respectively. The reason for this phenomenon is that the two communities are located outside of the commercial area of Hudai Town and are far from the Hudai Industrial Zone; thus, there is less infrastructure. Also, LiRen Garden#1 was the most occupied residential area, which resulted in a low SARSF-RA value. There were demolition and resettlement houses in LiRen Garden#1 and LiRen Garden#2, and the number of surrounding supporting facilities was small. However, the number of occupants was large, and it could be concluded that the comprehensive service facilities capacity of the demolition and resettlement housing community outside the commercial area was poor. Based on the existing analysis, the SARSF-RA largely depended on the hierarchical position, the maturity of the location, and the location of the community.

In additional, Graduated color symbology is used to show a quantitative difference that can classify the SARSF-RA in the experimental area, as shown in Figure A1c. The low-value area was interlaced with the high-value area. Also, it was block-shaped, and the spatial differentiation was evident, which indicated that the comprehensive serviceability was different at different levels and conditions. There were social phenomena that were similar to the classification and differentiation of communities. Some historical, social, economic, geographical, and administration factors have impacted urban planning orientation. From the perspective of the spatial allocation rationality of service facilities, we attempt to consider the future direction and suggestion of urban planning for the experimental area. We suggest that increasing the surrounding commonweal service facilities over low SARSF-RA residential areas, such as LiRen Garden#1 and LiRen Garden#2, can attract more residents.

## 4. Discussion

To intuitively show the differences in the SARSF-RA values, we constructed a four-quadrant map and collected a photo of the façade from some typical communities, as shown in Figure 7, which was expected to be more intuitive in terms of explaining the coordination between ASFC-RA and PDC-RA. The value of ASFC-RA was the highest, but the value of PDC-RA was in the middle in HuaHuiYuan; thus, it obtained a high degree of rationality. The value of ASFC-RA and the value of PDC-RA was the highest in Versailles Estate, indicating it was highly reasonable. In LiRen Garden#1, the value of ASFC-RA was lower, but the PDC-RA was the highest. Thus, it was less generous. In LiRen Garden#2, the value of ASFC-RA was the lowest, and the PDC-RA value was lower. Thus, LiRen Garden#2 obtained the most moderate degree of reasonableness. However, near the origin, the interrupted locating communities located in the fourth quadrant in Figure 7, including FuRunHuaYuan#1 and FuAnHuaYuan#3, had well-situated reasonableness, and the values of ASFC-RA and PDC-RA were more coordinated.

This paper studied the level of the service facilities and the actual population distribution in urban residential areas; additionally, the method used quickly, freely, and accurately obtained the relevant residential areas and auxiliary data for evaluation through open network data. Based on the population carrying capacity level in residential areas and the supply level of service resources, we constructed an indicator system of the comprehensive service level in residential areas, and we calculated the reasonableness of the service facilities in each district by the entropy weight method. The conclusions

are as follows: (1) Internet open data provide new verification data for service facilities rationality research at the micro scale, which has practical value for the evaluation of urban detailed planning and design. (2) It is found that the overall service resource supply level in the experimental area is higher in the northwest and lower in the southeast, and the communities with high and low levels show a robust spatial agglomeration phenomenon. The distribution of services in five major types also shows noticeable spatial agglomeration. This conclusion can help the government understand regional dominance; additionally, the population carrying capacity of the old community was generally higher than that of the new community, and the population carrying capacity near the community with a top service resource supply level was usually higher. This research has made significant progress in the application of big data, but the incompleteness of the data acquisition still limits it, and there are some shortcomings. The following points are worth further improvement and development: (1) The experimental area could be expanded to the whole city, and the scope of the experimental area can be enlarged in the later stage to obtain more stable law. (2) The number and scale of the service facilities are two important indicators for measuring the supply of the service facilities. Due to the lack of scale data, this study has certain limitations. It would also be worthwhile to study the method for supplementing the scale data to POI through other Internet datasets. (3) The spatial subdivision method of Tencent LBS data can improve to template-based GIS computation, such as a geometric algebra approach [35–41]. In this paper, the model provides a new and feasible Internet big data idea for the effective verification of residential area planning, which can be effectively applied to the evaluation of the service facilities in the micro-space residential areas. Additionally, the results can serve as a reference for city planning and decisions that support multi-source Internet data under a new trend.

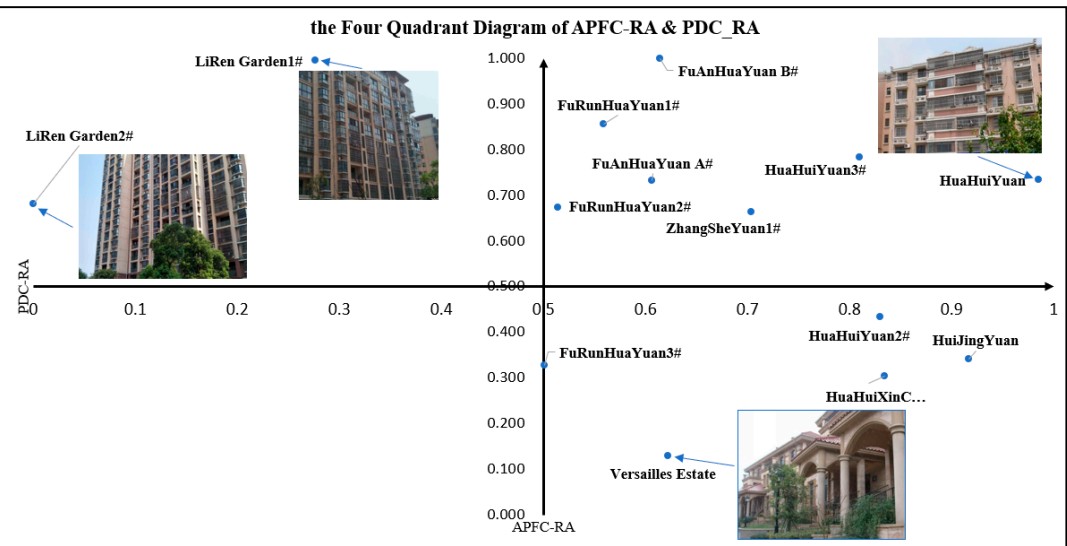

**Figure 7.** Four-quadrant coordination of ASFC-RA and population distribution capacity of residential areas (PDC-RA).

**Author Contributions:** Conceptualization, X.Z. and J.H.; Methodology, X.Z.; Software, X.Z. and C.H.; Validation, C.W., Y.D. and X.Z.; Data Curation, J.H.; Writing-Original Draft Preparation, X.Z.; Writing-Review & Editing, X.Z., Y.D.; Visualization, J.H.; Supervision, C.W.; Project Administration, C.W.; Funding Acquisition, C.W.

**Funding:** This research was funded by National Natural Science Foundation of China grant number 41471318.

**Conflicts of Interest:** The authors declare no conflict of interest.

## Appendix A

**Table A1.** Table of calculation results for each index in the experimental area.

| ID | Residential Area Name | $V_{ASFC-RA}$ | $V_{PDC-RA}$ | $V_{SARSF-RA}$ | Education Services | Employment Services | Leisure Services | Life Services | Medicine Services | Travel Services | Population Count | Building Area | Completion Year | Floors |
|---|---|---|---|---|---|---|---|---|---|---|---|---|---|---|
| 1 | LiRen Garden#2 | 0 | 0.680 | 0 | 22 | 1249 | 33 | 803 | 26 | 26 | 1976 | 43761 | 2012 | 18 |
| 2 | LiRen Garden#1 | 0.276 | 0.994 | 0.278 | 26 | 1411 | 35 | 1008 | 34 | 39 | 2888 | 365801 | 2010 | 18 |
| 3 | FuRunHuaYuan#1 | 0.559 | 0.857 | 0.652 | 27 | 1521 | 44 | 1170 | 41 | 59 | 2488 | 90012 | 2012 | 12 |
| 4 | FuAnHuaYuan #B | 0.614 | 1.000 | 0.614 | 29 | 1528 | 42 | 1314 | 43 | 60 | 2904 | 105389 | 2012 | 12 |
| 5 | FuAnHuaYuan #A | 0.606 | 0.733 | 0.827 | 30 | 1508 | 44 | 1290 | 40 | 57 | 2128 | 71925 | 2012 | 12 |
| 6 | HuaHuiYuan#3 | 0.81 | 0.782 | 1.035 | 31 | 1603 | 45 | 1392 | 44 | 100 | 2272 | 110903 | 2008 | 6 |
| 7 | HuaHuiYuan | 0.985 | 0.733 | 1.344 | 34 | 1608 | 48 | 1441 | 49 | 125 | 2129 | 145609 | 2008 | 6 |
| 8 | ZhangSheYuan#1 | 0.704 | 0.664 | 1.060 | 29 | 1554 | 44 | 1362 | 44 | 80 | 1928 | 69740 | 2012 | 12 |
| 9 | FuRunHuaYuan#3 | 0.501 | 0.328 | 1.528 | 26 | 1501 | 40 | 1273 | 40 | 57 | 952 | 43069 | 2012 | 12 |
| 10 | FuRunHuaYuan#2 | 0.514 | 0.672 | 0.765 | 26 | 1506 | 40 | 1289 | 41 | 57 | 1952 | 31860 | 2012 | 18 |
| 11 | HuaHuiYuan#2 | 0.83 | 0.433 | 1.919 | 33 | 1642 | 45 | 1311 | 42 | 103 | 1256 | 85329 | 2008 | 6 |
| 12 | HuiJingYuan | 0.917 | 0.342 | 2.684 | 34 | 1569 | 45 | 1424 | 49 | 117 | 992 | 40908 | 2009 | 6 |
| 13 | HuaHuiXinCun | 0.834 | 0.303 | 2.752 | 33 | 1572 | 45 | 1367 | 46 | 99 | 880 | 51238 | 2005 | 5 |
| 14 | Versailles Estate | 0.622 | 0.129 | 4.804 | 26 | 1529 | 42 | 1357 | 43 | 81 | 376 | 12711 | 2010 | 3 |

Note: The building average floors were obtained by field survey; the building area was obtained by multiplying the floors and the base map of the building based on the remote image from Tianditu; the completion year was obtained from the real estate agent, and the website address is https://wuxi.anjuke.com/sale/hudaizhen/.

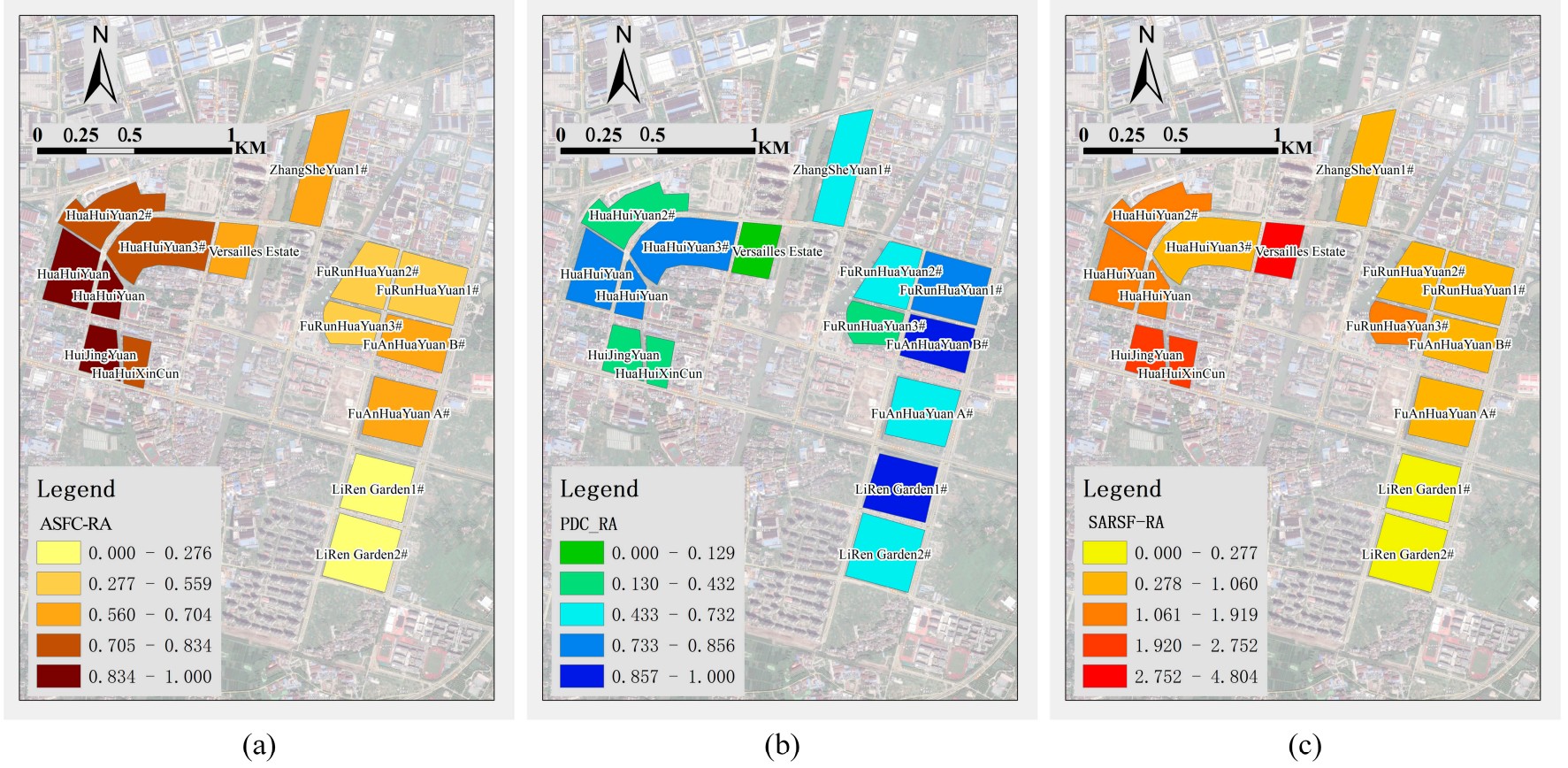

**Figure A1.** The spatial distribution of results.

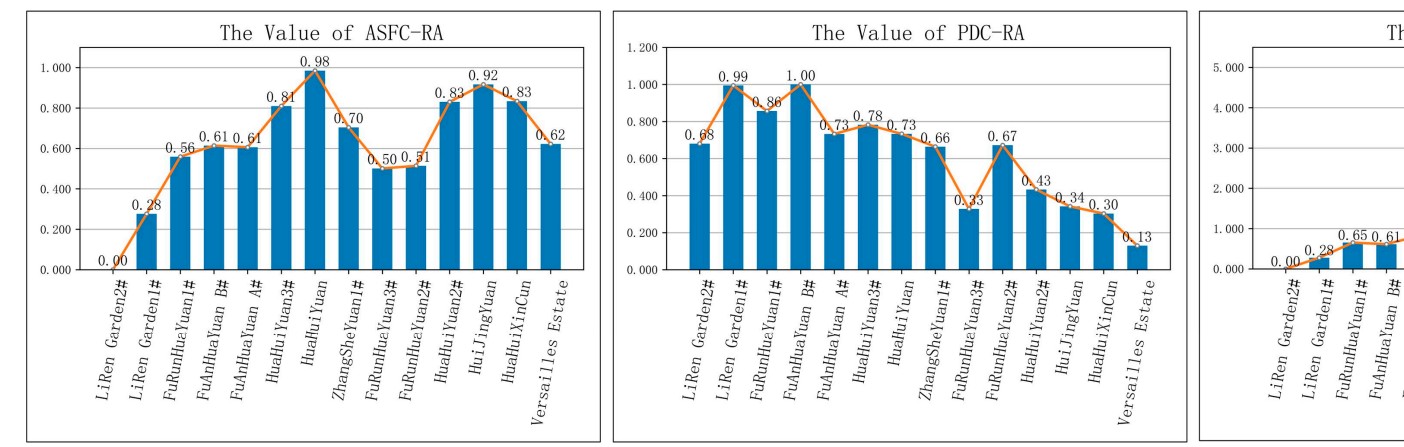

**Figure A2.** Bar charts of the result.

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
