# Peer review of "Measuring the Spatial Allocation Rationality of Service Facilities of Residential Areas Based on Internet Map and Location-Based Service Data"

_sustainability, doi:10.3390/su11051337_

Round 1
Reviewer 1 Report
This paper presents an interesting methodology regarding the use of internet maps for the study of the spatial allocation of service facilities in relation to the population allocation in residential areas. I believe that the paper can be further improved so as to become of a much wider interest. My main comments are the following:
a) The introduction of the paper should be enriched with a brief discussion on the relevant literature so as to understand better the contribution of the paper within this literature. An important part of the introduction as it stands now refers mainly to methodological issues.
b) In the presentation of the study area it is necessary to provide a brief description of the characteristics of the specific sections of this area because the references in specific names that are included in the discussion is not enough to understand the qualitative characteristics of these areas which are more important for the international reader.
c) In the discussion the authors should give more emphasis on what their results provide regarding the spatial allocation of the different service facilities in the different areas. It would be interesting here to mention also the urban planning aspects and discuss its relation to the findings of the spatial allocation of service facilities in residential areas.
d) A very wide polishing of the language is absolutely necessary. The paper will gain a lot from such a polishing as it is not that easy to read it in many parts.
e) Line 43-44: “Urban Residential Area Planning and Design Specification (GB50180-93): refer whose country is this guidance
f) Line 16: what does 'microcosmic knowledge' mean?
g) Accessibility: explain why accessibility is assessed only in travel cost and not in travel time
h) I don’t find necessary the use of the numbers (1), (2), (3) in the description of the types of data in the introduction (Lines 77, 79, 83).
Author Response
Thank you very much for your responsible review of my paper. I have got great gains in revise process.

Reviewer 2 Report
This article presents an interesting proposal to use Big Data to analyze the rationality of service facilities allocation in the micro-scale of a small part of the city. The authors have Identified new possibilities for analyzing data such as LBS, POI and data of route planning in Web-based mapping services. However, there are some shortcomings, both in developed methodology and in quality of presentation . This article require some improvements.
Quality of Presentation: acronyms, index and marks in Equation formulas, symbol explanations, names and descriptions of drawings and tables are inconsistent. For example, first we have ASFC-RA and at other times we have APSC-RA, etc. Explanations of symbols in Equation formulas are often not placed directly under them. All this confusion makes this article difficult to read. The bar charts presented at the figures are not histograms. Lack of consequences is also visible in the proposed methodology. For example, different normalization methods for different values are used without any justification. In the description of the methodology the normalization of ASFC-RA values is not mentioned only to appear in the chapter presenting the results. Map legends and descriptions of drawings are sometimes inconsistent. For example, in Figure 1, red points on the map legend have been described as points of Tencent LBS but in the Figure description they are described as a representation of the POIs distribution. However, diagrams shown in Figure 2 and Figure 4 are well-designed and facilitate general understanding of the idea.
Method: The are two issues that raise serious doubts in the described method of analysis:
1) The selection and classification of POI (points of interest). The authors determined the scale of their analysis as a micro scale - covering several close residential areas. On the one hand, the airports have been selected as POIs (they have large area of influence) and on the other hand, road signs and lights have been selected too (which have very small areas of influence). Analyzing them together, according to the presented methodology, seems to make little sense. The authors should very carefully chose POI categories which are proper for the scale of the analysis. The airport serves the population from large areas, and the neighborhood (proximity) of the airport is not necessarily something positive for communities due to significant noise problem. Some doubts are also related with the classification of "business residence" into "life" category, not into "employment" category, which requires an explanation.
2) The use of spatial statistic analysis for few objects. the authors conduct spatial autocorelation analysis for only 14 residential areas. The use of Moran's I spatial autocorelation analysis and HotSpot analysis (Getis-Ord Gi * statistics) for such a small number of objects is very problematic. These spatial autocorrelation analyzes contribute little to current research and they are problematic, so perhaps should be omit. The description of the results of this analysis also needs some improvements.
More detailed comments are following:
line 72: "internet maps" - does it man Web-based mapping services?
line 125: description of step 4 needs improvements. It is not sure what is the base for what.
line 124: VAPFC-AR or VASFC-AR ?
line 137 - acronyms problem again
line 161 - i, j, s,d - need descriptions
Formula (1) and (2) - problem with acronyms
Formula (4) - it is necessary to show what sumi(s) and sumi(all) means. It seems to be the same as VASFC-AR and VASFC-AR temp, respectively. If yes - why Authors provide new symbols?
line 173 - "relative entropy" is it Qi? Is "m" the number of facilities category?
Formula (6) - Value of Qi for every i-th residential area is constant. So it would be much simpler to calculate formula (6) in form : Qi * SUM[...]
line 195 - "time of life, education" - not clear meaning
Algorithm 1 - index m had different meaning in earlier formulas
line 231/232 - "Through the group statistic" It needs more explanation: what kind of, how?
line 246 - 257 - this paragraph needs some improvements/ clarifications
Formula (7) - marks and acronyms are inconsistent with the earlier formulas
line 263 - n? - probably m?
line 323-327 - the use of the word "ladder" makes the text difficult to understand
line 253 - "number of floor" - is it average or maximum number of floor of residential area buildings?
Author Response

(The authors gave the same response as above.)
